# A Paternal Methylation Error in the Congenital Hydrocephalic Texas (H-Tx) Rat Is Partially Rescued with Natural Folate Supplements

**DOI:** 10.3390/ijms24021638

**Published:** 2023-01-13

**Authors:** Naila Naz, Ghazaleh Moshkdanian, Salma Miyan, Sereen Eljabri, Charlotte James, Jaleel Miyan

**Affiliations:** Division of Neuroscience, Faculty of Biology, Medicine and Health, The University of Manchester, 3.540 Stopford Building, Oxford Road, Manchester M13 9PT, UK

**Keywords:** H-Tx rat, folate, ALDH1L1, hydrocephalus, FRα, FOLR1, brain, testis, sperm, methylation

## Abstract

Folate deficiencies, folate imbalance and associated abnormal methylation are associated with birth defects, developmental delays, neurological conditions and diseases. In the hydrocephalic Texas (H-Tx) rat, 10-formyl tetrahydrofolate dehydrogenase (FDH) is reduced or absent from the CSF and the nuclei of cells in the brain and liver and this is correlated with decreased DNA methylation. In the present study, we tested whether impaired folate metabolism or methylation exists in sexually mature, unaffected H-Tx rats, which may explain the propagation of hydrocephalus in their offspring. We compared normal Sprague Dawley (SD, *n* = 6) rats with untreated H-Tx (uH-Tx, *n* = 6 and folate-treated H-Tx (TrH-Tx, *n* = 4). Structural abnormalities were observed in the testis of uH-Tx rats, with decreased methylation, increased demethylation, and cell death, particularly of sperm. FDH and FRα protein expression was increased in uH-Tx males but not in folate-treated males but tissue folate levels were unchanged. 5-Methylcytosine was significantly reduced in untreated and partially restored in treated individuals, while 5-hydroxymethylcytosine was not significantly changed. Similarly, a decrease in DNA-methyltransferase-1 expression in uH-Tx rats was partially reversed with treatment. The data expose a significant germline methylation error in unaffected adult male H-Tx rats from which hydrocephalic offspring are obtained. Reduced methylation in the testis and sperm was partially recovered by treatment with folate supplements leading us to conclude that this neurological disorder may not be completely eradicated by maternal supplementation alone.

## 1. Introduction

DNA modifications are major epigenetic mechanisms to control the expression of multiple genes. One of the most studied of these epigenetic modifications occurs through an exceptionally coordinated mechanism that comprises de novo methylation, maintenance of the methylated state, and demethylation. A key tool in these studies has been the methylation of cytosines catalysed by DNA methyltransferase (DNMT) [1]. Methyl groups are supplied by s-adenosyl methionine (SAM), the universal methyl donor, which is generated through folate metabolism. Methylation happens at the fifth carbon position of a cytosine residue to form 5-methyl cytosine (5mC). Demethylation also occurs to form 5-hydroxy-methyl cytosine (5hmC), a product of 5mC demethylation catalysed by the Ten-Eleven Translocation (TET) protein family [1,2]. Demethylation has been reported to control many cellular and developmental processes, including the pluripotency of embryonic stem cells, differentiation and neuron development, and tumorigenesis in mammals, for example [3,4,5,6]. The factors that regulate the balance of DNA methylation/demethylation are not clearand represent a key target need to understand how these modifications are involved in both normal development and disease progression.

Folate is a regulating factor in the methylation cycle, and therefore, low levels of it may change the epigenetic controls of essential genes, particularly those involved in development, that are directed by DNA methylation [7,8,9,10,11]. Among the many neurological conditions affected by folate levels are neural tube defects and hydrocephalus (HC). HC is characterised by inflated brain ventricles due to excessive accumulation of cerebrospinal fluid (CSF) with raised intracranial pressure. Although the causes and mechanisms underlying the development of congenital HC are not thought to be the same as those involved in neural tube defects, folate and vitamin B_12_ do have fundamental roles in the functioning of the CNS and in the prevention of disorders that affect its development [12,13,14] including hydrocephalus [15,16,17,18]. To the present date, there is no prevention or cure for HC with clinical management only by neurosurgical CSF diversion using shunts or third ventriculostomy with associated high morbidity and shunt failure rates. Multiple genetic abnormalities have been associated with hydrocephalus [12,19,20,21,22,23,24,25] presenting a challenge to identify the cause. In our studies of this condition, we found a specific lack in the folate binding protein and enzyme, 10-formyl tetrahydrofolate dehydrogenase (FDH, also known as aldehyde dehydrogenase-1L1 (ALDH1L1)), in the CSF [15,18]. This was found in both affected H-Tx rats and in human neonates suffering congenital, postnatal infection-induced and post-haemorrhagic hydrocephalus, as well as in birth asphyxia (Naz, et al., in preparation). Significantly, susceptible H-Tx foetuses responded positively to natural folates given as maternal supplements but not to folic acid that actually precipitated hydrocephalus in them [18]. Our research thus indicates an important, if not vital role for the cerebrospinal fluid folate supply [15,16,17,18,26] for normal development and prevention of hydrocephalus.

The direct genetic cause of HC in humans is limited to the X-linked L1CAM gene that only affects boys and accounts for less than 1:1000 of total cases [27,28] while multiple other genes are associated with HC [22,25,29]. In the H-Tx rat, only one study has suggested a single autosomal recessive gene may be responsible for HC [30,31], while a number of publications by Hazel Jones and colleagues report that the hydrocephalus phenotype is controlled by a combination of multi-genetic and epigenetic factors [32,33,34,35,36,37,38,39,40]. However, the specifics of the genetic abnormalities causing congenital hydrocephalus in H-Tx rats remain unknown. In our previous publications, we reported that a deficiency in FDH entering the nucleus of brain and liver cells was linked to decreased DNA methylation, which could be a key factor in the developmental deficits associated with congenital and neonatal hydrocephalus [15,17]. DNA hypo-methylation can be reportedly overturned by folate supplementation [41,42,43,44] and the fact that we could reduce the risk of hydrocephalus in the H-Tx rat indicates that a potential methylation fault may be responsible [18].

Indeed, food fortification with folate in the USA and other countries has improved neurological deficits since its introduction in 1998 [45,46,47]; periconceptional folic acid supplementation remains the gold standard for optimal reductions in maternal folate insufficiencies that lead to a variety of conditions [48,49,50,51,52] including hydrocephalus [53,54]. Up to 30% of neural tube defects are not rescued by folate supplementation, perhaps needing inositol in addition to folate [55,56,57,58,59]. Little or no research has been carried out on the contribution of methylation and the potential beneficial effects of supplementation in both sexes. Food fortification supplements for both men and women in the USA are associated not only with a reduction in neural tube defects but also congenital heart defects, cardiovascular issues and certain cancers [60] indicating a potential paternal benefit of fortification including unexplained infertility [61]. Due to our finding that the risk of hydrocephalus in the H-Tx rat, with its unknown genetic cause, can be reduced with folate, we hypothesised that a methylation error in either or both parents may be responsible. In the present study we, therefore, aimed to understand if a folate-related fault in methylation in sexually mature male H-Tx rats could contribute to the risk of HC in susceptible foetuses and whether this risk could be rescued with folate supplements.

## 2. Results

### 2.1. Histological and Morphological Analysis of Testes

In SD rats, haematoxylin and eosin staining demonstrated a normal histological appearance of the testes (Figure 1a,b). The seminiferous tubules had a normal arrangement of spermatogonia (Sg) with an intact basement membrane. The conventional structure of primary spermatocytes (Ps), round (Rs) and elongated spermatids (Es) and mature sperms (Ms) were observed. By contrast, in H-Tx rats the testes appeared to have an abnormal morphology (Figure 1c,d). Degenerated seminiferous tubules were observed, with primary spermatogonia (Sg) located around distorted basement membranes. Vacuolated cytoplasm in round spermatozoa (Rs), and mature sperm with small dark nuclei and a distorted lumen (L) were observed. In SD rats, acridine orange staining indicated intact double-stranded DNA (stained green) in the sperm heads reflecting normal/healthy sperms (Figure 1e,f). However, in the H-Tx rats, the sperms were primarily stained red or orange, indicating a distorted/broken single-stranded DNA and possible cell death (Figure 1g,h). Overall, these results indicate a striking difference in testis and sperm morphology and viability between the SD and H-Tx rats. This is reflected in the poor fertility and survival of embryos in the H-Tx rat.

### 2.2. Changes in Folate Metabolism

Immunostaining indicates differential expression of FDH, folate receptor alpha (FRα), the main folate transporter, and folate (Fol) in the SD and H-Tx rat testes (Figure 2, Figure 3 and Figure 4) with variable expression for all three target molecules both between and within different tubules. It is possible this is related to the localised needs for folate in cell division and the production of sperm. FDH was in Leydig cells and spermatogonia, while in the secondary spermatozoa, round cells, elongated sperm and mature sperm it was less abundant. In all FDH was mainly cytoplasmic with nuclear localisation in SD and TrH-Tx testes indicating a potential need for FDH in the process of DNA methylation (white arrow in Figure 2A,B). The cellular localisation is more apparent in the peroxidase staining where the white arrows indicate nuclear localisation in SD and TrH-Tx and the black arrow indicates the absence of nuclear expression in uH-Tx. Western blot analyses of tissue lysates (Figure 2B,C) show a significant increase in FDH in uH-Tx compared to SD or TrH-Tx (27,660 vs. 20,437 OD, *p* = 0.0069, and 19,493 vs. 27,660, *p* = 0.0243, respectively).

FRα (also called folate receptor 1, FOLR1) was observed in Leydig cells and spermatogonia, with less abundance in secondary spermatozoa, and round cells. Elongated and mature sperm had abundant FRα in SD and TrH-Tx but in uH-Tx. Localisation was seen in some elongated sperm in some tubules and greatly reduced in other cells with most FRα staining localised outside the tubules (Figure 3A). In the untreated H-Tx rats, FRα was expressed abundantly in patches and clusters of cells around the basal membrane, localised in the cytoplasm by contrast to SD and TrH-Tx which had more nuclear localisation (Figure 3A(b,d,f)). TrH-Tx showed similar staining to SD. Immunoperoxidase staining (Figure 3B) shows concentrated FRα outside the tubules and in Leydig cells with barely any nuclear stain compared with the more intense staining in uH-Tx including nuclear staining. Western blot analyses of tissue lysates show significantly elevated FRα in uH-Tx compared to SD (OD: 25,952 and 17,868, respectively, *p* = 0.0029) with no significant difference between SD and TrH-Tx expression (Figure 3B,C).

Immunostaining for folate shows concentrations of folate in the nuclei of primary spermatozoa in SD and TrH-Tx testes (Figure 4A(a,b,e,f)). In uH-Tx testes, folate was in Leydig cells, and nuclei of spermatogonia, spermatozoa and some sperm heads with primary spermatozoa in far fewer numbers than in either SD or TrH-Tx (Figure 4A(c,d)). Sperm heads are more numerous and folate positive in TrH-Tx than in SD or uH-Tx. The cellular localisation is more apparent in the peroxidase staining (Figure 4B) where the arrows show both nuclear and cytoplasmic expression in both SD and TrH-Tx. In uH-Tx staining was more intense and in most cells, including smooth muscle. Dot blot analysis indicated a non-significant decrease in folate concentration in both uH-Tx and TrH-Tx compared to SD. Taken together, the reduced nuclear FDH expression, and increased nuclear FRα and folate in untreated H-Tx testes suggested more folate availability in the nucleus but a potential lack of use of available folate due to reduced FDH.

### 2.3. Changes in Methylation

SD rats show abundant expression of 5-methyl cytosine (5mc) in the cells around the basal membrane and most other cells including mature sperms (yellow and white respectively in Figure 5A(a)) which is also seen with peroxidase staining (red and black arrows respectively in Figure 5A(b)) indicating high levels of DNA methylation. The primary and secondary spermatocytes and round spermatids show variable expression of 5mc (blue arrow in Figure 5A(b)). Reflecting this are low levels of demethylation indicated by low expression of 5-hydroxymethyl cytosine (5hmc, yellow arrow in Figure 5A(g), black arrow in 5A(j)) and also found in some mature sperm in peroxidase staining (red arrow in Figure 5A(j)). In uH-Tx, much less 5mc was observed in the testes as a whole with positive staining restricted to Leydig cells (yellow arrow in Figure 5A(c), black arrow in 5A(d)) and some mature sperm (white arrow in Figure 5A(c), red arrow in 5A(d)). There was barely detectable 5mc in spermatogonia (blue arrows in Figure 5A(c,d)) and sperm stained by immunoperoxidase (red arrow in Figure 5A(d)). Treated H-Tx (TrH-Tx) show better than SD control levels of methylation with increased expression of 5mc in Leydig cells, primary spermatozoa (yellow and blue arrows in Figure 5A(e), black arrow in Figure 5A(f)), secondary spermatozoa and mature sperm (blue and red arrows in Figure 5A(e,f)). In this case the improvement is clearer in the immunofluorescence-stained sections (Figure 5A(e)) although mature sperm cannot be detected except in peroxidase staining. There is decreased demethylation associated with the improved methylation in both SD and TrH-Tx compared to uH-Tx (Figure 5A(g,h,i), and Figure 5A(j,l,n) respectively). Dot blot analysis (Figure 5B) confirmed a significant reduction in 5mc in uH-Tx testes compared to SD controls (Figure 5C, mean OD: 12,198 and 23,950, respectively, *p* = 0.0013), which was raised in TrH-Tx though this was not significant on whole tissue lysate analysis compared to the evidence from immunostaining. Dot blots also confirmed increased 5hmc in uH-Tx compared to both SD and TrH-Tx (Figure 5D, mean OD: 27,927, 20,014 and 18,243, respectively, *p* = 0.049). Associated with the decreased methylation in uH-Tx was a significant reduction in s-adenosyl methionine (SAM), the universal methyl donor, compared to both SD and TrH-Tx (Figure 5E, mean OD: 10,006, 17,220 and 13,686, respectively, *p*: 0.0001).

Unsurprisingly, with high levels of methylation, SD rats had an abundant expression of DNA methyl transferase-1 (DNMT-1) in the cells around the basal lamina, including Leydig cells, primary and secondary spermatocytes, round spermatids and mature sperms (Figure 5A(k)). By contrast, DNMT-1 was reduced in uH-Tx (Figure 5A(m)) but with positive staining mainly in cells around the basal lamina and primary spermatogonia and with more mature cells lacking expression. TrH-Tx had improved DNMT-1 expression resembling that seen in SD rats (Figure 5A(o)). Associated with decreased methylation in uH-Tx, dot blot analysis also showed a decrease in S-Adenosyl methionine (SAM), the universal methyl donor involved in methylation, in uH-Tx which was increased after treatment (Figure 5E). Altogether, these results indicate a decrease in methylation and increased demethylation in the H-Tx rat testis as compared to SD rats, with decreased DNMT-1 and SAM, which was partially restored by folate treatment.

## 3. Discussion

The current study reports for the first time male-related errors in folate metabolism and methylation that could underlie hydrocephalus and its inheritance in the H-Tx rat. The H-Tx rat recapitulates the clinical signs and symptoms of human congenital hydrocephalus in many aspects [62,63] with grossly enlarged cerebral ventricles apparent in late gestation, between embryonic day 17 to 18 [64,65,66,67], and with a complex mode of inheritance giving an incidence of 30–50% affected pups [32,65]. No H-Tx foetus is “normal”, with all having susceptibility triggered by maternal stress and nutritional and environmental factors. Mating unaffected individuals maintains the inheritance, confirming the fact that none are normal and carry affected genes.

The results of the current study showed differences in the size, shape and abnormality in the tubular arrangement within the untreated H-Tx testes. Abnormal testicular morphology was characteristic of H-Tx rats with a normal regular pattern in the arrangement and shape of seminiferous tubules some with intact basal laminae and others with incomplete laminae. Moreover, an observed decreased cell proliferation, altered testicular histology and increased mature cell death in the H-Tx testes must contribute to adverse reproductive outcomes, including offspring with hydrocephalus and other malformations that are embryonic lethal, as many dead or reabsorbed embryos are found in the uterus of pregnant dams. Similar adverse reproductive outcomes, including altered testicular histology, smaller testis sizes, and lower sperm counts have been reported in C57BL/6 methylene tetrahydrofolate reductase (MTHFR)-deficient mice indicating a direct folate link [68].

Our findings of hypo-methylation (decreased 5-methyl cytosine, 5mc) and hyper-demethylation (increased 5-hydroxy-methyl cytosine, 5hmc) in combination with decreased DNMT-1 indicate defective methylation in the testes and sperm of untreated H-Tx rats, which was partially recovered with folate treatment (Figure 6). The low levels of FDH, FRα and folate we found in the cells, and particularly in the nuclei of uH-Tx, would also result in failure of methylation. The increased demethylation may be a physiological consequence of low folate and FDH as well as DNMT1. This could be investigated by looking at levels of TET enzymes. The increased FDH, FRα and folate, found in western and dot blots of total tissue lysates, and in the testes indicate a good supply to the organs but a failure in delivery and transfer into cells, and nuclei suggesting a concentration due to lack of use. The failure of entry into the nucleus is particularly interesting as it suggests some active barrier to entry to the biological molecules which can be circumvented by alternative folate specieis given as supplements, similar to our findings for hydrocephalus. Defective DNA methylation in germ cells can have profound effects through altered gene expression. Poor DNA methylation, caused by a mutation in the methyl-CpG-binding protein 2 [69,70] or impaired DNMT activity in the adult brain [71] is linked to a number of diseases and neurological disorders [72,73]. A reduction of only 8% in global DNA methylation levels is associated with neurodegeneration [71] perhaps explaining the significant populationg of dead cells in the untreated H-Tx testes. An imbalance between methylation and demethylation is also thought to be responsible for problems with learning and memory and developmental abnormalities including autism and dementia [69,74,75]. Abnormalities in DNMT enzymes, which are responsible for the methylation of DNA, result in altered genomic methylation in germ cells and infertility [61,76]. All of these studies as well as our reported findings indicate that methylation and the required supply of folate to make this successful, is critical to successful pregnancies, normal development and function of the individual, particularly the brain, for the lifespan.

Studies in humans have shown altered methylation patterns in the mature sperm have the potential of being transmitted and may adversely affect progeny outcome while mice deficient in DNA methyltransferase (DNMT1) do not survive gestation [77,78,79,80,81,82,83,84,85]. This fits with what we find in the H-Tx rat where we have reduced fertility, failed embryonic development and hydrocephalus in surviving offspring associated with the male sperm methylation error. With methylation reactions essential for cell division, the synthesis of membrane phospholipids, myelin basic protein and neurotransmitters, for example, it is not surprising to find a significant methylation error in the H-Tx rat associated with poor outcomes in pregnancy since the maintenance of normal DNA methylation patterns is essential for normal gene expression, genomic imprinting, and cellular differentiation, as already discussed.

Overall, our data suggest that the male H-Tx reproductive system is folate compromised during pre-puberty, which has been reported to cause lifelong changes in methylation status as well as in outcomes of offspring. The data suggest impaired folate metabolism and obstructed methylation with hyper-demethylation in the testes of untreated H-Tx rats. Treatment with a bioactive folate combination of 5-formyl tetrahydrofolate and tetrahydrofolate one month before sexual activity partially reversed the methylation deficit throughout the tissue compared to untreated rats. Bioactive folate is already known to be essential for DNA methylation where folic acid, a synthetic form, is not useful [86,87,88] and we have demonstrated that it can significantly reduce the risk for hydrocephalus in the H-Tx rats [18].

### Limitations of the Current Study and Future Work

Although western and dot blots provided semi-quantitative analysis of proteins, we did not carry out quantification of immunofluorescence which therefore provide qualitative comparisons only. A larger number of samples would make quantification more beneficial. This study was carried out on a Japanese strain of the H-Tx rat recovered from frozen embryos. This then needs to be compared to the European and American strains as it is possible that different rates of inbreeding, nutrition and environmental factors may have induced different levels of these methylation errors, in addition to the effects of long-term storage of frozen embryos. Further studies need to be performed to establish the severity of methylation impact on these rats. Moreover, a more prolonged folate treatment could possibly offer long-term benefits in reversing the apparently inherited methylation errors. Studies in humans may reveal paternal methylation errors associated with hydrocephalus as they have for other conditions discussed in the text.

## 4. Materials and Methods

### 4.1. Animals

All experiments were sanctioned by the Home Office Animals (Scientific Procedures) Act Inspectorate (UK) and were carried out under project licence PPL70/8025. Colonies of SD (control group) and H-Tx rats (experimental group) were kept on a 12 h light 12 h dark cycle commencing at 8 am, at a constant temperature, humidity and filtered air, with free access to food and water and low light and sound levels. The H-Tx colony was maintained through brother-sister mating between unaffected animals, and the SD colony was maintained through random pair mating. The animals were fed the standard Beekay rat and mouse diet no. 2 (B and K Universal, Hull, UK). H-Tx foetuses were categorised as either affected or unaffected H-Tx based on the excessive CSF accumulation which showed as a gross dooming of the head of affected individuals under a Leica MZ6 microscope (Milton Keynes, UK). Sexually mature unaffected H-Tx and SD males were used for the study.

#### 4.1.1. Animal Treatment

In this study, we used three groups of animals. Sexually mature male SD as healthy controls, sexually mature male untreated H-Tx and sexually mature treated male H-Tx. Treatment was started 28 days prior to timed mating. Animals were randomly assigned to the treatment groups to receive either: Saline (0.9% NaCl, *n* = 7) or a combination of tetrahydrofolic acid (Sigma-Aldrich, Gillingham, UK) and 5-formyltetrahydrofolic acid (folinic acid, Alfa Aesar, Heysham, UK) at a dose of 2.25 mg/kg each (*n* = 7). All doses were administered through the subcutaneous route to bypass any gut malabsorption and to ensure full dose delivery.

#### 4.1.2. Animal Collection

Adult rats were euthanised by intraperitoneal injection of sodium Pentobarbitone 20% *w*/*v* (Pentoject from Animalcare Ltd., York, UK). Testes were collected and one of each from each male was preserved in paraformaldehyde while the other was immediately frozen with dry ice-cooled isopentane (VWR International Ltd., Lutterworth, UK) and stored at −80 °C.

### 4.2. Immunohistochemical Analysis

#### 4.2.1. Antibodies Used in the Study

Primary antibodies used were anti-rabbit ALDH1L1 at 1:2000 dilution and rabbit anti-FRα at 1:10,000 (Proteintech, Manchester, UK), mouse anti-folic acid (Sigma Aldrich, Gillingha, UK) at 1:2000, mouse anti-5mc at 1:500, and mouse anti-5hmc (Abcam, Cambridge, UK), rabbit anti-DNMT1 (Epigentek, Farmingdale, NY, USA) at 1:500, mouse anti-beta-actin (Cell Signalling Techology, Leiden, The Netherlands) at 1:10,000. Secondary antibodies were goat anti-rabbit HRP and goat anti-mouse HRP (Cell Signalling Technology, Leiden, The Netherlands ), or goat anti-rabbit Coralite 488 or 594 and goat anti-mouse Coralite 488 or 594 (all from Proteintech, Manchester, UK). Alternatively, Alexa Fluor-conjugated secondary antibodies were also used at 555, 488, Cy7 (Abcam, Cambridge, UK).

#### 4.2.2. Tissue Preparation

For best tissue morphology, paraformaldehyde-fixed tissues were used to obtain cryostat sections. After fixation, tissues were incubated in 30% sucrose until they sunk. The tissues were then snap-frozen in dry ice cooled isopentane, mounted on a chuck and located in a Leica CM1900 cryostat. Sections were cut at 25 µm thickness and collected onto Superfrost Plus adhesion slides (Medline Scientific Ltd., Chalgrove, UK). Slides were dried and kept at −20 until further use.

### 4.3. Haematoxylin and Eosin Staining

Sections were incubated in Y-eosin 0.5% aqueous solution (Sigma Aldrich, Gillingham, UK) for 5 min, rinsed with 1% PBS and the nuclei then counter-stained with haematoxylin solution (Sigma Aldrich, Gillingham, UK) for 30 s. Slides were washed with water to remove any excess stain and then mounted with gelatine (Sigma Aldrich, Gillingham, UK).

### 4.4. Acridine Orange Staining

Sections were incubated in 0.1 M HCl at room temperature for 1 min. Slides were then incubated in acridine orange staining solution (Sigma, Gillingham, UK) for up to 5 min followed by a quick rinse in PBS. Sections were mounted with VECTASHIELD^®^ Antifade Mounting Media (Vector laboratories, Burlingame, CA, USA).

### 4.5. Peroxidase Staining

Peroxidase staining was performed according to manufacturer instructions using DAKO EnVision+ System, Peroxidase (Agilent Technologies LDA, Stockport, UK). Negative controls were obtained by incubating with isotype-specific mouse/rabbit/goat IgGs instead of the specific primary antibody. Nuclei were counterstained with Meyer’s hemalum solution (Sigma Aldrich, Gillingham, UK) before the slides were mounted with coverslips.

### 4.6. Immunofluorescence Staining

Cryostat sections were obtained as described above. Blocking of non-specific protein binding was performed utilising goat serum for 1 h at room temperature. The slides were incubated overnight at 4 °C with the primary antibody diluted in 5% Triton X-100 and (3%) BSA-PBS buffer. Antigens were visualised using immunofluorescence Coralite and Alexa Fluor conjugated secondary antibodies: Goat anti-rabbit 488/594, Goat anti-rabbit 549/555, Goat anti-mouse Cy 7. Negative-control immunostaining was performed by omitting the primary antibody and using an isotype-matching control immunoglobulin. Slides were mounted with VECTASHIELD^®^ Antifade Mounting Media (Vector laboratories, Burlingame, CA, USA). Sections were scanned on a 3D Histech Pannoramic 250 Flash Slide Scanner (3DHistech Ltd., Budapest, Hungary) using a 20×/0.80 Plan Apochromat objective (Zeiss, Jena, Germany) before viewing and obtaining snapshots in 3D Histech Caseviewer software. No quantitative analysis was carried out of staining intensity or co-localisation in this initial study as the n number was too small for statistical interpretations. Qualitative differences are highlighted between the tissues examined as a target for more extensive and detailed, quantitative studies.

### 4.7. Total Protein Isolation

Total protein was isolated from fresh frozen testes using RIPA buffer (Sigma Aldirch, Gillingham, UK) with 1× Roche complete protease inhibitor cocktail (Sigma Aldrich, Gillingham, UK) added. Tissues were homogenised using a Polytron homogenizer (at 3–4 speed) and centrifuged at 13,000 rpm for 20 min. The supernatant was collected into another Eppendorf tube and centrifuged again to remove any debris. All steps were carried out at 4 °C. Proteins were stored at −20 °C.

### 4.8. Protein Determination and Western Blots

Protein concentration was determined with a Nanodrop instrument (Fisher Scientific, Loughborough, UK). For SDS PAGE electrophoresis a concentration of 200 µg of protein was loaded into each well of a 4–12% (*w*/*v*) polyacrylamide 10 or 12 well NuPAGE mini-gel (Fisher Scientific, Loughborough, UK). Protein semi-dry transfer was carried out using the iBlot Transfer system and iBlot gel transfer stacks with nitrocellulose (Fisher Scientific, Loughborough, UK) with a 7 min transfer time. The nitrocellulose membranes were stained with a few drops of Ponceau solution (Sigma Aldrich, Gillingham, UK) for 15–30 s to test that protein transfer had been successful. Membranes were washed twice for 3 min each time with 1× PBS. To perform Western blots, membranes were blocked with 5% BSA prepared in 1× PBS; with 1% Triton X added, for 1 h at RT. The primary antibodies were diluted in the blocking buffer described above. Primary antibody incubation took place overnight at 4 °C, and membrane incubations were performed on a rotating shaker. The membranes were washed four times for 15 min each time with 1× PBS-Triton X and placed on a rotating shaker, at room temperature. Secondary antibodies diluted in blocking buffer were incubated at room temperature for 1 h. The membrane was washed four times for 15 min each time with 1× PBS-T. For HRP secondaries, an Amersham ECL kit (Fisher Scientific, Loughborough, UK) was used to visualise labelled bands. For fluorescent secondaries, membranes were visualised directly. Results were obtained using the ChemiDoc instrument (BioRad laboratories Ltd., Watford, UK) and were recorded and analysed using Image Lab 4.1 (BioRad Laboratories Ltd., Watford, UK).

### 4.9. Dot Blots

Dot blots were performed to evaluate changes in metabolites. Each sample was diluted to a final concentration of 200 µg/mL and 5 µL were pipetted onto separate pre-determined locations of a nitrocellulose membrane strip and allowed to air dry. Membranes were then processed for respective antibodies as mentioned in western blot sections.

### 4.10. Statistical Analysis

Six SD control, six untreated H-Tx and four treated H-Tx rats were used in this study. One testis from each rat was cut into sections and at least three slides containing at least three sections each were analysed using immunohistochemistry, giving a total of nine sections per rat; 54 SD, 54 untreated H-Tx and 36 treated H-Tx sections were analysed for each target molecule/antibody. Frozen testes were used for protein extraction and analysis. Blots were repeated at least three times for each target molecule/antibody and used as technical replicates. Due to the low n number, we assumed normality for biological analysis and used two-tailed unpaired Student’s t-tests to compare the data from the three groups. Results from Western and dot blots were analysed using Image Studio 5.2 (LI-COR Biotechnology Ltd., Cambridge, UK) or Image Lab 6.0 (Bio-Rad Laboratories Ltd., Watford, UK) to record intensities and total signal. All data were transferred into GraphPad Prism 9.0 (Boston, MA, USA) for analysis and graphing.

## Figures and Tables

**Figure 1 ijms-24-01638-f001:**
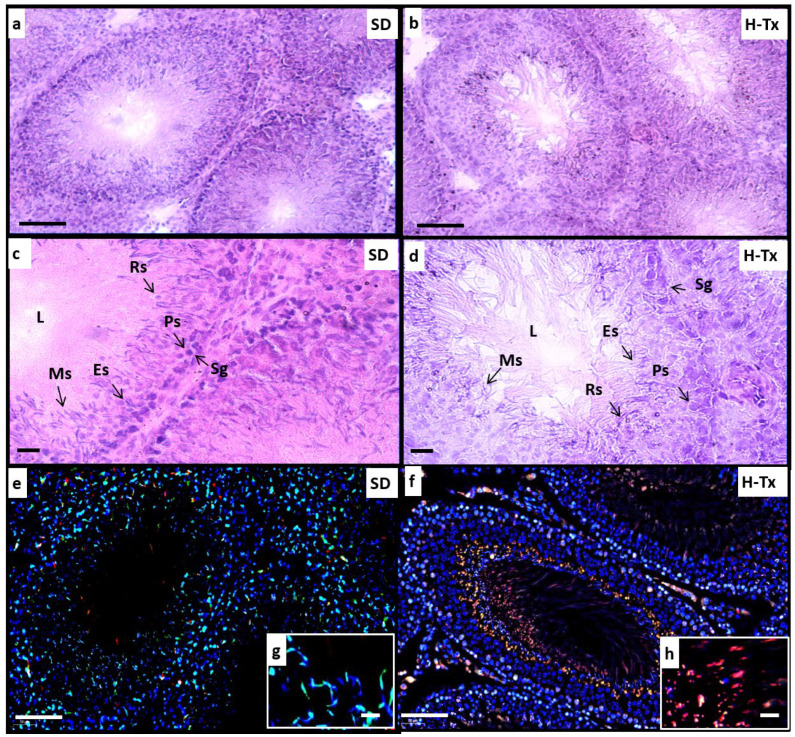
Histochemistry of rat testes. (**a**–**d**) Haematoxylin and eosin staining to demonstrate histological structure of rat testes. SD and untreated H-Tx rats ((**a,b**) at 200×, scale: 50 µm and (**c,d**) at 400×, scale: 25 µm) demonstrate normal appearance of seminiferous tubules with normal arrangement of spermatogonia (Sg) resting on an intact basement membrane (arrowhead). Normal structure of primary spermatocytes (Ps), round (RS) and elongated spermatids (ES) and mature sperms (Ms). Untreated H-Tx rats (**b**,**d**) had degenerated seminiferous tubules, and primary spermatogonia (Sg) were distributed around distorted basement membranes (arrowhead). Numerous vacuolated cytoplasm of round spermatozoa (RS), sperm with small dark nuclei and a distorted lumen (L). Acridine orange staining (**e**,**f**) of SD rat testes (**e**) gives a green stain indicating cells/sperm with intact and double-stranded DNA. In H-Tx rat testes (**f**) red stain indicates distorted, single-stranded DNA. Orange and yellow stain represents cell death (magnification (**e**,**f**): 200×, scale: 50 µm). Inset (**g**,**h**) show sperm at higher resolution to illustrate dead and fragmented cells in the H-Tx (**h**) compared to the intact sperm in SD (**g**) (magnification (**g**,**h**): 800×, scale: 12.5 µm). The results are representative of *n* = 4–6 rats with a minimum of three sections from different locations of testes.

**Figure 2 ijms-24-01638-f002:**
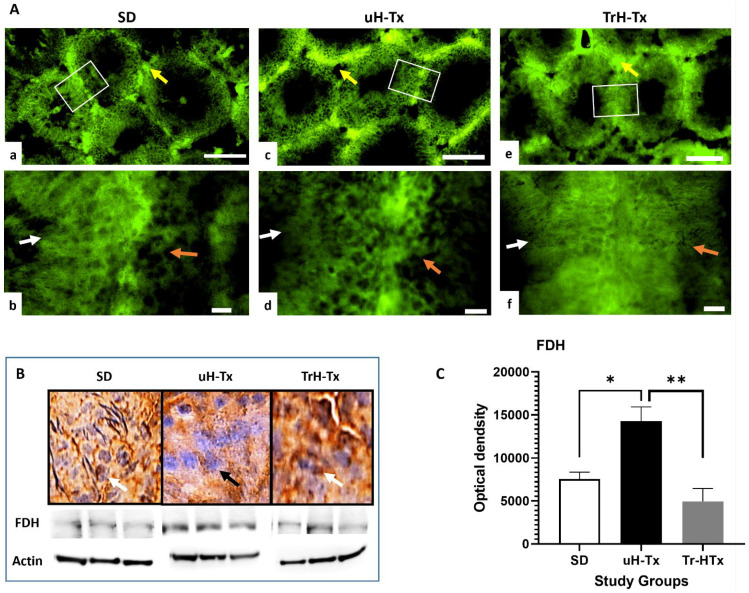
FDH expression in rat testes. (**A**) Immunofluorescence staining for FDH in SD (**a**,**b**), untreated H-Tx (uH-Tx) (**c**,**d**) and treated H-Tx (TrH-Tx) (**e**,**f**) rats ((**a**,**c**,**e**) at 100×, scale bar = 200 µm and (**b**,**d**,**f**) at 400×, scale: 25 µm). White boxes in (**a**,**c**,**e**) are regions enlarged in (**b**,**d**,**f**). Yellow arrows in (**a**,**c**,**e**) indicate intertubulular tissue regions with concentrations of FDH. FDH shows a variable pattern between different tubules in all three groups with cytoplasmic localisation (white and orange arrows indicate tubule tissue-lumen margins in adjacent tubules). (**B**). Immunoperoxidase staining shows nuclear expression of FDH in SD rat testes (white arrow point to positively stained nuclei in both SD and TrH-Tx) whereas, uH-Tx has little or no nuclear localisation (black arrow points to negatively stained nuclei). TrH-Tx shows similar staining to SD (magnification 400×, scale: 25 µm). Western blot analysis of total tissue lysate shows increased FDH protein in uH-Tx compared to both SD and TrH-Tx, shown in graphical format in (**C**) (ODs SD: 20,437, H-Tx: 27,660, TrH-Tx: 19,493, *: *p* ≥ 0.05, **: *p* ≥ 0.01). The results are representative of *n* = 4–6 rats with a minimum of three sections from different locations of each testis and three repeated blots.

**Figure 3 ijms-24-01638-f003:**
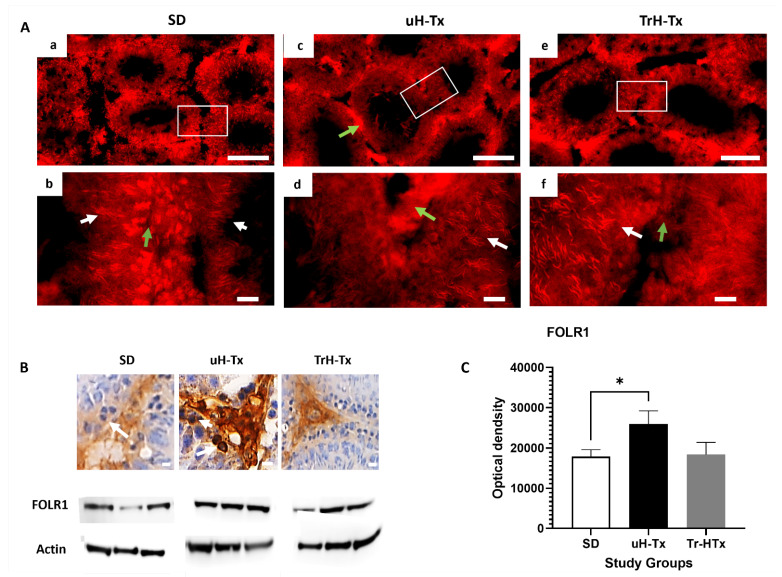
Folate receptor alpha (FRα or FOLR1) expression. (**A**) Immunofluorescent staining for FOLR1 in SD (**a**,**b**), uH-Tx (**c**,**d**), and TrH-Tx (**e**,**f**) ((**a**,**c**,**e**) at 100×, scale: 200 µm and (**b**,**d**,**f**) at 400×, scale: 25 µm). White boxes in (**a**,**c**,**e**) are regions enlarged in (**b**,**d**,**f**). Green arrow points to areas outside tubules, including smooth muscle, that are FOLR1 positive in uH-Tx but not in SD or TrH-Tx, with little if any positive staining within uH-Tx tubules except for mature sperm (white arrows) (**B**) Immunoperoxidase staining of FOLR1 more clearly demonstrates the difference in subcellular localisation. White arrows indicate nuclear FOLR1 expression in uH-Tx testes and a lack of nuclear expression in SD and TrH-Tx rats (400×, scale: 25 µm). Western blot analysis of tissue lysates (**B**) shows an increased FOLR1 protein expression in uH-Tx with restoration to SD control levels in TrH-Tx rats (**C**). Densitometric analysis shows statistically significant increased FOLR1 in uH-Tx testes compared to SD (*: *p* ≥ 0.05). The results are representative of *n* = 4–6 rats with a minimum of three sections from different locations of testes and three repeated blots.

**Figure 4 ijms-24-01638-f004:**
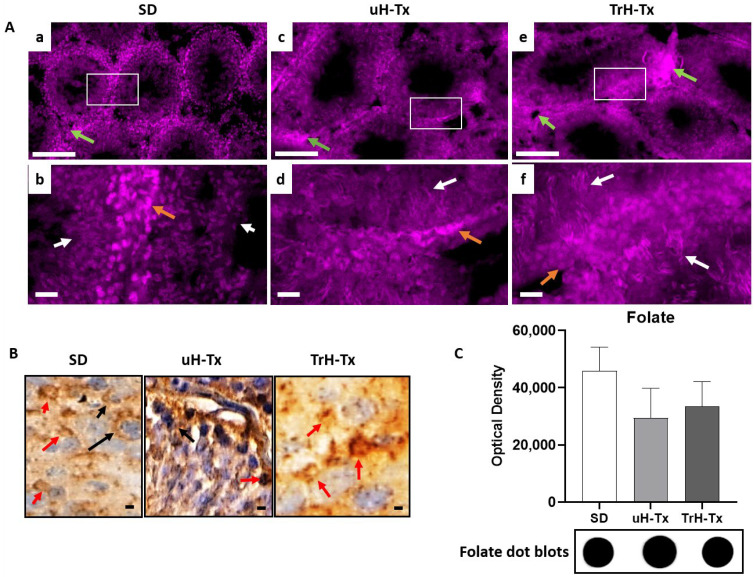
Folate in rat testes. (**A**) Immunofluorescent staining for folate in SD (**a**,**b**), uH-Tx (**c**,**d**) and TrH-Tx (**e**,**f**) ((**a**,**c**,**e**) at 100×, scale: 200 µm and (**b**,**d**,**f**) at 400×, scale: 25 µm). White boxes in (**a**,**c**,**e**) are regions enlarged in (**b**,**d**,**f**). Green arrows point to smooth muscle and other tissue outside tubules while orange arrows point to primary spermatogonia in SD and TrH-Tx (**b**,**f**) and Leydig cells in uH-Tx (**c**,**d**). White arrows point to areas of mature sperm that are not clearly visible in SD and are more numerous and more intense for folate in TrH-Tx. (**B**) Immunoperoxidase staining for folate shows the difference in subcellular localisation. Black arrows indicate the membrane/cytoplasmic folate localisation in SD while red arrows indicate nuclear localisation in uH-Tx rats that are recovered to SD control patterns in TrH-Tx. There are some scattered cells in SD and TrH-Tx that do have nuclear localisation (black arrows). Magnification 400×, scale: 25 µm. (**C**) Dot blot analysis of total tissue lysate shows a non-significant decrease in folate in uH-Tx and TrH-Tx testes. The results are representative of *n* = 4–6 rats with a minimum of three sections from different locations of testes and three repeated blots.

**Figure 5 ijms-24-01638-f005:**
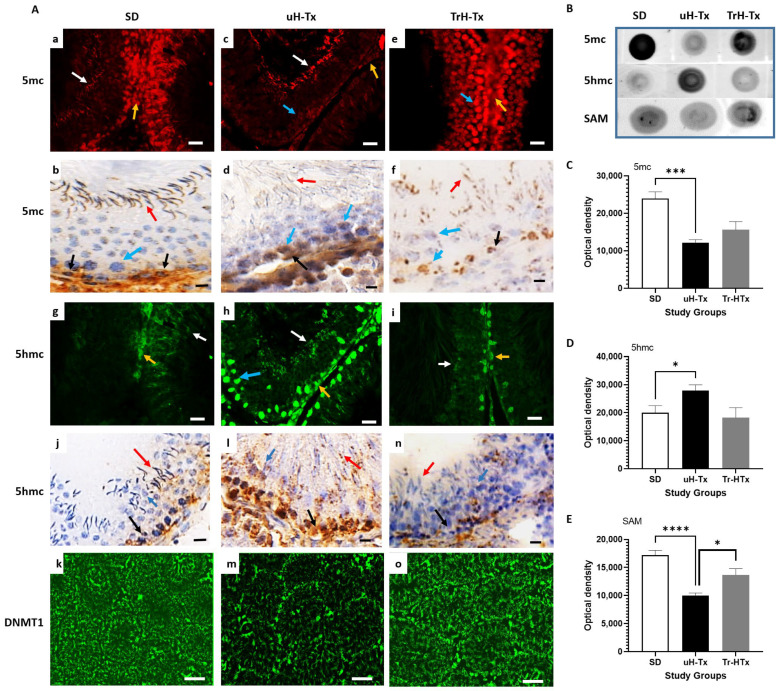
Changes in methylation in rat testis. (**A**) Immunostaining for 5 methyl cytosine (5mc) in SD (**a**,**b**) uH-Tx (**c**,**d**) and TrH-Tx (**e**,**f**); 5 hydroxymethyl-cytosine (5hmc) in SD (**g**,**j**) uH-Tx (**h**,**l**) and TrH-Tx (**i**,**n**) and for DNMT1 in SD (**k**), uH-Tx (**m**) and TrH-Tx (**o**). Yellow (**a**,**c**,**e**) and black arrows (**b**,**d**,**f**): Leydig cells, blue arrows (**b**,**c**,**d**,**f**,**h**): spermatocytes, white (**a**,**c**,**e**) and red (**b**,**d**,**f**) arrows: mature spermatozoa. Yellow arrows in (**g**,**h**,**i**) and black arrows in (**j**,**l**,**n**) indicate spermatogonia and/or primary spermatocytes. White arrows in (**g**,**h**,**i**) and red arrows in (**j**,**l**,**n**) indicate mature sperm. Magnification 200×, scale: 50 µm. (**B**) Dot blot analysis of total tissue lysate stained and analysed for (**C**) 5mc, (**D**) 5hmc and (**E**) SAM. Results were statistically significant for 5mc (***, *p* = 0.0003), 5hmc (*, *p* = 0.049) and SAM (****, *p* = 0.0001). The results are representative of *n* = 4–6 rats, with a minimum of three sections from different locations of testes, and three repeated blots.

**Figure 6 ijms-24-01638-f006:**
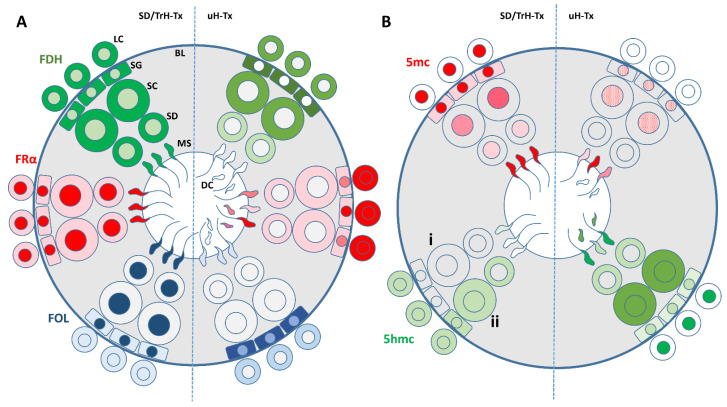
Summary diagrams of the main findings from this study. (**A**) Diagram showing localisation and relative concentrations of FDH (green), FRα (red) and folate (FOL) (blue) in SD controls and treated H-Tx (SD/TrH-Tx—**left** hemisphere) compared with untreated H-Tx (uH-Tx—**right** hemisphere). All cells contain FDH in the cytoplasm and nucleus in SD and TrH-Tx. In uH-Tx all cells except mature sperm contain FDH in the cytoplasm, at higher levels in spermatogonia (SG) and Spermatocytes (SC) but less in Spermatids (SD). Mature sperm (MS) in uHT-x contain no FDH and many dead and fragmented sperm (DC) are found in the lumen of the tubules. FRα (red) is mainly localised in nuclei including sperm heads with lower levels in the cytoplasm of SD and TrH-Tx. In uH-Tx, FRα fills Leydig cells (LG) but is present at low levels in the cytoplasm of spermatocytes and spermatids with no expression in nuclei except in spermatogonia and some sperm heads including fragmented sperm. In SD and TrH-Tx, folate (FOL) (blue) is concentrated in nuclei with low expression in the cytoplasm of spermatogonia but not in the cytoplasm of other cells except Leydig cells that have low levels in cytoplasm and nuclei. In uH-Tx, folate is concentrated in the cytoplasm and nuclei of spermatogonia but in no other cells except very low levels in sperm heads and in the cytoplasm of Leydig cells. (**B**) Diagram showing the methylation status, measured as 5-methyl cytosine (5mc), and demethylation status, measured as 5-hydroxymethyl cytosine (5hmc), in SD controls and treated H-Tx (SD/TrH-Tx—**left** hemisphere) compared with untreated H-Tx (uH-Tx—**right** hemisphere) associated with the findings in (**A**). All cell nuclei are methylated in SD and TrH-Tx with some staining in the spermatogonia cytoplasm, presumably associated with high levels of RNA. In uH-Tx low levels of methylation are found in spermatogonia and spermatocytes but none is seen in spermatids although there are some in some mature sperm and dead fragments. The marker of demethylation, 5-hydroxymethyl cytosine (5hmc) is not present in any cells except Leydig cells in SD controls (i) but in TrH-Tx (ii) all cells have low levels in cytoplasm and nuclei but not in the nuclei of spermatids or sperm heads. In uH-Tx, 5hmc is at high levels in the cytoplasm and nuclei of spermatocytes, lower levels in spermatogonia and only in the cytoplasm of spermatids and some high levels in sperm heads and fragments. Low levels of FDH, FRα and folate in the cells, particularly in the nuclei of uH-Tx, would result in failure of methylation. The increased demethylation may be a physiological consequence of low folate and FDH as well as DNMT1. The increased FDH, FRα and folate, found in Western and dot blots of total tissue lysates, in the testes indicates a good supply to the organs but a failure in delivery and transfer into cells suggesting a concentration due to lack of use.

## Data Availability

Data can be obtained through reasonable request to the corresponding author.

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
