# Peer review of "A Paternal Methylation Error in the Congenital Hydrocephalic Texas (H-Tx) Rat Is Partially Rescued with Natural Folate Supplements"

_ijms, 2023, doi:10.3390/ijms24021638_

Round 1
Reviewer 1 Report
The Manuscript by Naz et al. (Manuscript ID: ijms-2041025) on " A paternal methylation error in the congenital hydrocephalic Texas (H-Tx) rat is partially rescued with natural folate supplements " aimed to understand if a folate-related fault in methylation in sexually mature rats can have a connection with impaired methylation contributing to HC in susceptible fetuses. The authors compared normal SD rats with untreated H-Tx and folate-treated H-Tx rats. The authors observed differences in treated and untreated rats. Moreover, authors observed significant germline methylation error in unaffected adult male H-Tx rats from which hydrocephalic offspring are obtained. After treatment with folate supplements partial recovery in reduced methylation was also observed in the testis and sperm. Authors think that the neurological disorder may not be completely eradicated by maternal supplementation alone. The work is interesting. I have some comments that may help to improve the quality of the manuscript as follows:
- There are several spelling errors in the manuscript. For example, on page 2, line 89, defict should be deficit or defect. On page 3, line 105, impared should be impaired. Thus, the manuscript should be checked for English language, grammatical, and spelling corrections.
- Authors should provide high-magnification images for Immunofluorescent staining for FDH (Figure 2A) and folate receptor alpha, FOLR1 (Figure 3A).
- To understand the overall conclusion of the manuscript, authors should add a schematic representation.
- The dot blot analysis of total tissue lysate presented in Figure 5B needs to be repeated along with the quantification since the quality of dot blot is not good.
Author Response
We thank the reviewer for their positive comments and suggestions for improvements.
1.We have revised the entire manuscript and corrected all spelling and grammatical errors we could find and have had it read by colleagues to ensure we captured the maximum number of issues.
2.All figures have been revised with higher power images in the second rows to give much clearer details.
3. a major summary figure has been added to the discussion section.
4.The dot blots have been repeated and replaced.
Reviewer 2 Report
In this paper by Naila Naz et al. entitled “A paternal methylation error in the congenital hydrocephalic Texas (H-Tx) rat is partially rescued with natural folate supplements“ authors aimed to test whether impaired folate metabolism or methylation exists in sexually mature, unaffected H-Tx rats, that may explain the propagation of hydrocephalus in their offspring.
Overall, the topic is important and intriguing, the reference list is updated and cover the relevant literature adequately, there are some issues to be addressed.
1. I would recommend the authors make the introduction section more specific and clear, finalize the aim (make links to your previous research in this area).
2. In the description of the results, please specify the quantative data (not just significance levels). If no quantitative assessment has been made in immunofluorescence, then it makes sense to make a separate section on the limitations of the study.
3. In the list of references, remove duplicate numbers.
4. The article lacks a graphical representation of the study design.
5. In the discussion, it makes sense to summarize all the data obtained in the scheme of pathogenesis, this will simplify the understanding of the article.
6. In the statistical analysis section, the question of the normality of the distribution remains. According to the description the authors used 4-6 animals per group, please specify how it was statistically analysed.
Author Response
We thank the reviewer for his constructive comments. We hope we have adequately addressed all the issues raised.
- We have revised the introduction and the rest of the manuscript to address the issues.
- We have now included the data in the description of the results and have noted in the methods why quantitative analysis was not carried on the imaging studies due to low n numbers. Limitations to the study have been added as a consequence.
- References have been corrected.
- A detailed summary diagram and legend has been added to the discussion.
- We hope the discussion is now clearer with reference to the summary diagram.
- The statistical analysis section has added detail on the design and analysis used.